Subject Category:
Biology (whole organism)

Subject Areas:
behaviour/physiology/environmental science

Keywords:
$CO_2$, mollusc, carbon dioxide, climate change

Author for correspondence:
Rachael M. Heuer
e-mail: rheuer@rsmas.miami.edu

# Ocean acidification affects acid–base physiology and behaviour in a model invertebrate, the California sea hare (Aplysia californica)

## Rebecca L. Zlatkin and Rachael M. Heuer

University of Miami Rosenstiel School of Marine and Atmospheric Science, Department of Marine Biology and Ecology, 4600 Rickenbacker Causeway, Miami, FL 33149, USA

 RLZ, 0000-0002-2426-0814; RMH, 0000-0002-8327-9224

Behavioural impairment following exposure to ocean acidification-relevant $CO_2$ levels has been noted in a broad array of taxa. The underlying cause of these disruptions is thought to stem from alterations of ion gradients ($HCO_3^-/Cl^-$) across neuronal cell membranes that occur as a consequence of maintaining pH homeostasis via the accumulation of $HCO_3^-$. While behavioural impacts are widely documented, few studies have measured acid–base parameters in species showing behavioural disruptions. In addition, current studies examining mechanisms lack resolution in targeting specific neural pathways corresponding to a given behaviour. With these considerations in mind, acid–base parameters and behaviour were measured in a model organism used for decades as a research model to study learning, the California sea hare (Aplysia californica). Aplysia exposed to elevated $CO_2$ increased haemolymph $HCO_3^-$, achieving full and partial pH compensation at 1200 and 3000 µatm $CO_2$, respectively. Increased $CO_2$ did not affect self-righting behaviour. In contrast, both levels of elevated $CO_2$ reduced the time of the tail-withdrawal reflex, suggesting a reduction in antipredator response. Overall, these results confirm that Aplysia are promising models to examine mechanisms underlying $CO_2$-induced behavioural disruptions since they regulate $HCO_3^-$ and have behaviours linked to neural networks amenable to electrophysiological testing.

## 1. Background

Ocean acidification is occurring at rates not observed in the last 300 million years. Average global oceanic $CO_2$ levels are projected to

increase from current levels of approximately 400 to approximately 940 µatm $CO_2$ by the end of the century and approximately 1900 µatm $CO_2$ by the year 2300 unless the rate of $CO_2$ emissions is substantially curtailed [1–3]. This rapid rate of change has made predicting the sensitivity of organisms to future predicted $CO_2$ levels a major focus of climate change research. Early studies focused heavily on calcifying invertebrates, reporting widespread impacts to calcification and growth [4]. Fish exposed to $CO_2$ have exhibited alterations to mitochondrial pathways, intestinal base secretion and otolith growth [5–7].

In addition, impaired behaviour following $CO_2$ exposure has been reported in more than approximately 130 studies to date in marine organisms at ocean acidification-relevant $CO_2$ levels (less than 1900 µatm $CO_2$). The majority of these studies have focused on marine fish, noting impairments to various endpoints including vision, olfaction, lateralization and learning [8–11], reviewed in [12]. Examination of behavioural disturbances has also been extended to invertebrates, where negative effects on predator defence behaviours have been observed [13–16].

The underlying cause of these behavioural disruptions is thought to result from the compensatory mechanism that allows fish and some active invertebrates to maintain pH homeostasis when exposed to elevated $CO_2$. Following the onset of $CO_2$ exposure, animals that are acid–base 'regulators' counter an initial drop in blood pH through the retention and/or uptake of $HCO_3^-$. This process allows acid–base regulators to correct pH to pre-exposure levels; however, both $HCO_3^-$ and $PCO_2$ remain elevated [12,17,18]. This compensation mechanism is generally related to how 'active' an organism is, as higher metabolic rates ($O_2$ consumption) necessitate higher rates of $CO_2$ excretion [19]. The accumulation of $HCO_3^-$ in extracellular fluids is usually coupled with an equimolar decrease in $Cl^-$ [18,20]. The resulting changes in extracellular and intracellular $HCO_3^-$ and $Cl^-$ are thought to alter behaviour by attenuating the movement of these ions through the primary receptor ($GABA_A$) responsible for background inhibitory responses in the vertebrate and invertebrate nervous system [11,21,22]. Thus, strong acid–base regulators with the ability to accumulate $HCO_3^-$ are hypothesized to be most at risk for behavioural disturbances [11].

Nilsson and colleagues [11] first implicated $GABA_A$ receptor involvement in behavioural disruptions by treating $CO_2$-impaired animals with gabazine, a $GABA_A$ receptor antagonist. This treatment was found to reverse $CO_2$-induced behavioural changes. Similar subsequent studies have continued to provide evidence for the involvement of $GABA_A$ receptors using gabazine or muscimol ($GABA_A$ receptor agonist) in fish [8,10,23–28] and in some invertebrates [13,29]. While this methodology has been seminal in providing a parsimonious explanation for altered behaviour and $GABA_A$-receptor involvement in $CO_2$-induced disruptions, future studies would benefit from two important considerations. First, although this proposed mechanism hinges on changes that occur following $CO_2$ compensation, few studies have measured acid–base parameters in a marine species while also measuring behaviour [30–35]. Such measurements would be especially important in invertebrates, where there is more inherent variation in acid–base regulatory ability [19,36,37]. For example, sea urchins (*Paracentrotus lividus*) retain $HCO_3^-$ to defend pH, while mussels (*Mytilus edulis*) do not, and experience an acidosis when exposed to the same $CO_2$ level (1480 µatm $CO_2$) [38]. Second, although crucial in implicating the $GABA_A$ receptor, immersing an animal in seawater containing a $GABA_A$ receptor pharmacological agent lacks resolution in targeting specific behaviours and could induce effects on unintended targets [23]. In addition, there has been little exploration of potential alternative or additional mechanisms in $CO_2$-induced behavioural disruptions [39,40]. Finally, although the majority of $CO_2$ behavioural studies are performed on fish, the vertebrate nervous system is complex, making it difficult to link a particular behaviour to specific neural networks.

To address these limitations, we propose that future research examining the behavioural impacts of $CO_2$ would benefit from identifying a model organism well-suited for both acid–base balance and neurophysiological studies. The ideal study organism would meet three criteria: (1) a simple and well-mapped nervous system, (2) reproducible behavioural assays, and (3) an acid–base 'regulator' profile, with the ability to accumulate $HCO_3^-$ to defend pH. The California sea hare (*Aplysia californica*), referred to herein as 'Aplysia', is widely known to meet the first two criteria perfectly and has been used for decades as a biomedical research model to study the cellular basis of learning [41].

Since the ability to acid–base regulate has been linked to behavioural disruptions, measuring the baseline $CO_2$ acid–base response in Aplysia is a necessary step in assessing their feasibility as a model for $CO_2$ behavioural research. The first objective of the present study was to examine acid–base parameters in haemolymph from Aplysia exposed to either control (approx. 400), 1200 or 3000 µatm $CO_2$. Since Aplysia are not sessile invertebrates, it was hypothesized that they would exhibit an acid–base 'regulator' profile, and actively retain $HCO_3^-$ to defend pH following $CO_2$ exposure. The second objective of this study was to examine the impacts of elevated $CO_2$ on two simple behaviours with

well-characterized neural networks [42], the righting reflex and the tail-withdrawal reflex. Righting is important for orientation and reattachment to substrate, while the tail-withdrawal reflex is an antipredator response that activates muscles used in escape responses [42–44]. Elevated $CO_2$ was expected to alter behaviour, as noted in previous studies. Notably, the chosen behavioural assays and $CO_2$ levels are environmentally relevant for Aplysia living in the intertidal zone of the North American Pacific coast [45]. Ultimately, this study marks the first step in assessing Aplysia as a potential model for future studies of $CO_2$-induced behavioural disruptions in marine organisms, including exploration of the $GABA_A$ hypothesis in addition to potential alternative mechanisms.

# 2. Material and methods

## 2.1. Animal care and experimental exposure

Aplysia (*Aplysia californica*), hatchery-reared from egg masses of wild-caught animals, were provided by the National Resource for Aplysia (National Institute of Health Grant P40OD010952) at the University of Miami Rosenstiel School of Marine and Atmospheric Science. Prior to use in experiments, Aplysia were fed ad libitum with red macroalga *Gracilaria ferox* and *Agardhiella subulata* [46] and were kept in 16 l tanks with a seawater flow rate of approximately 1.3 l min$^{-1}$ at approximately 15°C.

During experimentation, Aplysia were acclimated to either control (400), 1200 or 3000 µatm $CO_2$ for acid–base ($n = 2–3$ tank replicates, 3–5 animals/tank) and behavioural experiments ($n = 4–7$ tanks, 2–5 animals/tank). These acclimations were performed in 16 l tanks with flow-through seawater (0.6 l min$^{-1}$, 15°C). Animals were exposed for either 4 or 11 days to each $CO_2$ level. Since day of exposure (4 versus 11) did not significantly impact any measured endpoint (see below), exposure duration is referred to as 4–11 days throughout the manuscript. These time periods have previously been sufficient to reach a stable $HCO_3^-$ accumulation for $CO_2$ compensation [47]. In addition, 4 days is close to the 5-day exposure period previously demonstrated to induce behavioural disruptions in other invertebrates [13,14]. Animals were permitted to feed on the first day of the exposure but food was subsequently withheld approximately 96 h prior to experimental testing. Animals that experienced 11-day exposures were subjected to the same approximately 96 h fasting period. Animals remained immersed in seawater throughout the experimental period. Animals used in experiments were approximately 10–11 months of age and weighed 90–110 g (electronic supplementary material, table S1).

## 2.2. Seawater $CO_2$ manipulation

Desired $PCO_2$ levels were achieved using a $CO_2$ negative feedback system as previously described (Loligo Systems, Denmark) [6]. First, a standard curve was made by determining the relationship between known gas standards and seawater pH. Using this relationship, a pH set-point corresponding to each desired $PCO_2$ level was calculated, and 100% $CO_2$ was slowly bubbled into flow-through, aerated tanks to achieve the chosen $PCO_2$ level. The pH electrode and meter (WTW Sentix H electrode and 3310 meter) corresponding to each experimental tank were connected to CapCTRL software that delivered $CO_2$ using solenoid valves controlled by a DAQ-M digital relay instrument (Loligo Systems). Validation of desired $PCO_2$ values was achieved using $pH_{NBS}$ and total $CO_2$ ($TCO_2$) and was performed approximately two times per experiment. Measurements of $pH_{NBS}$ were recorded multiple times per week using an independent pH electrode and meter (Radiometer PHC3005 electrode, ThermoFisher Orion Star A221 meter). A Corning 965 $CO_2$ analyser (Corning Diagnostics) was used to measure $TCO_2$. To calculate $PCO_2$ and titratable alkalinity (TA), values of $pH_{NBS}$ and $TCO_2$, were entered into CO2SYS [48]. Calculated $PCO_2$ values for control, 1200 and 3000 µatm $CO_2$ are reported in electronic supplementary material, table S2. Temperature and salinity were measured approximately three times per week (WTW 3310 meter and TetraCon 325; electronic supplementary material, table S2).

## 2.3. Objective 1: Haemolymph acid–base balance and ion measurements

Extracellular haemolymph was sampled by inserting a 500 µl gas-tight glass syringe (Hamilton) towards the posterior and alongside the foot of the animal and gently withdrawing fluid. Haemolymph was measured immediately for extracellular pH ($pH_e$) using a custom glass chamber fitted around a needle pH microsensor attached to pH-1 Micro meter (Loligo Systems). The pH microsensors were pre-calibrated from the manufacturer and were corrected after verification with a known $pH_{NBS}$ value

from sterile seawater. This sterile seawater was used to flush out the pH chamber in between sample measurements and was measured using Radiometer PHC3005 pH electrode attached to a ThermoFisher Orion Star A221 meter. Haemolymph from the same animal was measured for $TCO_2$ (Corning 965, Corning Diagnostics). $HCO_3^-$ and $PCO_2$ were calculated from $TCO_2$ and pH using the Henderson–Hasselbach equation using an established solubility constant ($\alpha CO_2$) and dissociation constant (pK) for carbonic acid [49].

## 2.4. Objective 2: General behavioural assay protocols

For all behavioural assays, animals were gently removed from their experimental tank and placed carefully in the bottom of test tanks in water at their respective acclimation $PCO_2$ level. Tanks were 16 l and had a depth of 16 cm. In all assays, animals were given a 5 min acclimation time to become accustomed to the test tank prior to commencing behavioural tests. All assays were recorded on video and the experimenter was blind to the experimental treatment both during experiments and video analyses. In some cases, animals were tested in one of the two behaviour assays on the 4th day of exposure, returned to acclimation tanks, then tested on the 11th day for the second behaviour. The order of behaviours tested was altered. Based on previous studies, even repeated stimuli or noxious stimuli do not elicit long-term memory formation (animals retested on day 7) [50–53]. Accordingly, there was no reason to suspect that the mild stimulus in the present study would impact animals receiving a second behavioural test. In both assays, animals remaining in a contracted state for more than 1 min or animals that inked during tests were eliminated from analyses as 'non-participators'. In previous studies, extreme stress has been shown to lead to tachycardia and suppressed reflex activity [54], and inking is considered a 'high-threshold, all or none' behaviour [55]. This criteria resulted in removal of five control, seven 1200 µatm and eight 3000 µatm animals from the tail-withdrawal assay. One animal was removed from the 1200 µatm and the 3000 µatm treatments during the righting assay.

## 2.5. Righting behavioural assay

Protocols followed those outlined in a previous study [42]. Following the 5 min acclimation period, the animal was gently lifted to the top of the water column and released while on its side. The start time of the reflex occurred the moment the animal made contact with the bottom of the tank. The time from bottom contact to when the animal returned to an upright position and initiated its first crawling was recorded as righting time. The assay was performed in triplicate with a rest period of 5 min between trials [42,43]. The data was summarized for each individual as the mean of the triplicate measurements for the reflex time.

## 2.6. Tail-withdrawal behavioural assay

Protocols followed those outlined in previous studies [42,56]. Following the acclimation period, the animal was carefully lifted off the test tank bottom, and gently held by the experimenter as close to the tank bottom as possible without allowing the animal to adhere to the bottom (approx. 1 cm). At this point, the length from the tip of the tail to the top of the head was measured and recorded as the resting length, using a transparent ruler lying next to the animal in the bottom of the tank. The animal was then placed on the bottom of the tank and a blunted 20G needle was pressed onto the tip of the animal's tail (approx. 50–70° angle) for one second to depress the tissue against the test tank bottom to a depth approximately half the thickness of the tail. This depression caused the tail to withdraw and represented the starting time of the reflex. At maximal contraction, the total length of the animal from the tip of the tail to the top of the head was noted using the ruler. Relaxation of the tail to approximately 50% of the original tail length signified the end of the reflex. The reflex was measured in triplicate with rest intervals of 10 min between each trial [42].

## 2.7. Statistical analysis

Linear mixed effect (LME) models were used to test for responses to $CO_2$ exposure levels for the time to complete the righting reflex and the time to complete the tail-withdrawal reflex. These models included $CO_2$ level and day of exposure as fixed factors, and tank as a random factor. Tukey's *post hoc* tests with a Holm-adjusted *p*-value was used to compare means between $CO_2$ exposure levels. The righting time and the tail-withdrawal reflex time data were log-transformed prior to analysis. A general linear model was

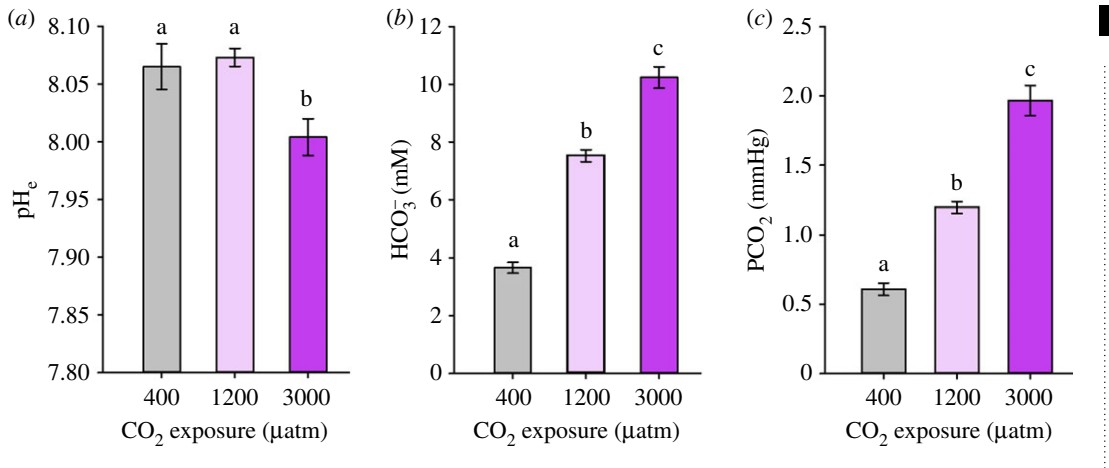

**Figure 1.** Haemolymph (*a*) $pH_e$, (*b*) $HCO_3^-$ and (*c*) $PCO_2$ in Aplysia (*Aplysia californica*) exposed to either control (400), 1200 µatm $CO_2$ or 3000 µatm $CO_2$ for 4–11 days. Values are reported as means ± s.e.m; $n = 10$–11. Means that share the same letter are not significantly different ($p < 0.05$).

used in instances where inclusion of tank as a random factor in mixed models resulted in overfit, using treatment and day as fixed factors. This applied to the per cent of the tail withdrawn in the tail-withdrawal reflex, haemolymph $pH_e$, haemolymph $HCO_3^-$ and haemolymph $PCO_2$. The per cent of the tail withdrawn in the tail-withdrawal reflex was arcsin transformed prior to statistical analysis. All models were conducted in R v. 3.5.2 [57] using the lme4 and lmerTest packages [58,59], and *post hoc* testing was conducted using the multcomp package [60]. Significance was determined at $p < 0.05$ for all tests and all values are presented as means ± s.e.m. Figures were made using SigmaPlot 13.0 and presented as treatment means pooled across days of exposure, since day of exposure was not significant in any test.

## 3. Results

### 3.1. Physiological measurements

For all parameters, the day of testing was not significant, so results are presented across $CO_2$ treatments. Haemolymph $pH_e$ was significantly affected by $CO_2$ exposure (figure 1*a*; $F_{2,29} = 5.94$, $p = 0.007$), but was not affected by the day of testing ($F_{1,29} = 0.248$, $p = 0.622$). Accordingly, *post hoc* comparisons on the effect of $CO_2$ exposure on haemolymph $pH_e$ reflected pooled values across days of exposure. Aplysia exposed to $CO_2$ for 4–11 days showed a significant reduction in haemolymph $pH_e$ at 3000 ($t = -2.736$, $p = 0.021$), but not at 1200 µatm $CO_2$ when compared with controls (figure 1*a*; $t = 0.373$, $p = 0.712$). As expected, Aplysia showed evidence of a compensatory response via the accumulation of $HCO_3^-$ (figure 1*b*). $HCO_3^-$ was significantly affected by $CO_2$ exposure ($F_{2,27} = 157.53$, $p < 0.001$), but was not affected by day of testing ($F_{1,27} = 2.15$, $p = 0.15$). *Post hoc* comparisons revealed significant differences in $HCO_3^-$ between all three $CO_2$ levels (figure 1*b*, all $p < 0.001$). $pCO_2$ increased significantly with $CO_2$ exposure ($F_{2,27} = 87.41$, $p < 0.001$), but was also not affected by day of testing ($F_{1,27} = 0.036$, $p = 0.850$). *Post hoc* testing revealed significant differences in $pCO_2$ between all three $CO_2$ levels (figure 1*c*, all $p < 0.001$).

The relationship between haemolymph $HCO_3^-$ and $PCO_2$ exposure was not perfectly linear, which probably accounts for incomplete pH compensation at 3000 µatm $CO_2$ (electronic supplementary material, figure S1).

### 3.2. Behavioural responses

Aplysia exposed to $CO_2$ displayed no difference in the time to right when compared with control animals (figure 2*a*; $F_{1,13} = 0.411$, $p = 0.533$). The day of testing did not affect the righting response ($F_{1,12} = 0.002$, $p = 0.964$). Tail-withdrawal time was significantly affected by increased $CO_2$ exposure ($F_{2,15} = 4.52$, $p = 0.029$), but was not affected by the day of testing ($F_{1,16} = 0.04$, $p = 0.84$). Accordingly, *post hoc* comparisons on the effect of $CO_2$ exposure on tail-withdrawal reflex time reflected pooled values across days of exposure. Animals exposed to 1200 and 3000 µatm $CO_2$ relaxed their tail approximately 36–37% faster than

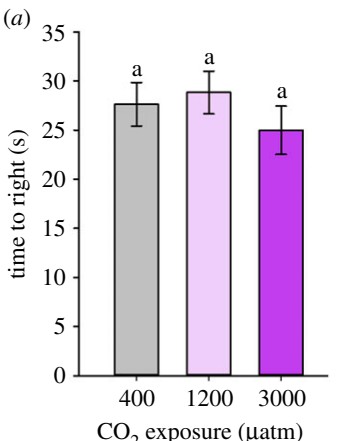
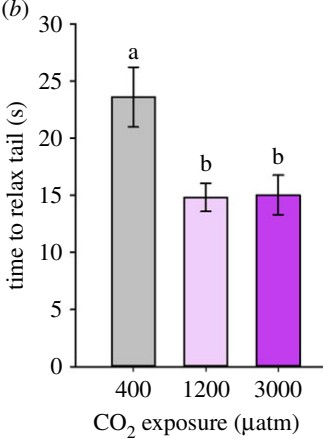
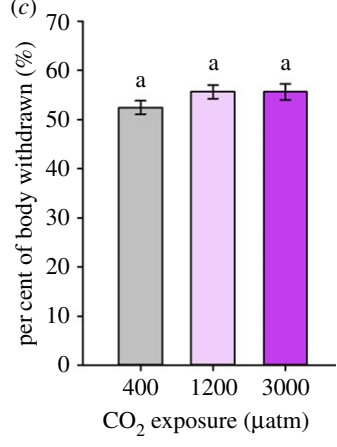

**Figure 2.** Behavioural analysis in Aplysia (*Aplysia californica*) exposed to either control (400), 1200 µatm $CO_2$ or 3000 µatm $CO_2$ for 4–11 days. (*a*) Righting reflex ($n = 13–16$), (*b*) tail-withdrawal reflex (TWR) amount of time to relax the tail to 50% of original length and (*c*) TWR percentage of starting body length withdrawn following tail touch ($n = 19, 17, 16$ for control, 1200 µatm $CO_2$ and 3000 µatm $CO_2$, respectively). All values are reported as means ± s.e.m. Means that share the same letter are not significantly different ($p < 0.05$).

control animals (figure 2*b*; $z = -2.521$, $p = 0.027$, $z = -2.612$, $p = 0.027$, respectively). High $CO_2$-exposed groups did not show a significant difference from one another ($z = 0.134$, $p = 0.893$). The percentage of body length withdrawn following tail depression exhibited no significant differences with treatment or day (figure 2*c*; treatment: $F_{1,49} = 2.58$, $p = 0.12$, day: $F_{1,49} = 1.36$, $p = 0.25$).

## 4. Discussion and conclusion

Aplysia exposed to elevated $CO_2$ (1200 and 3000 µatm $CO_2$) were able to accumulate significantly higher levels of $HCO_3^-$ in haemolymph following a 4–11 day exposure period (figure 1*b*). This compensatory effort led to complete pH defence at 1200 µatm $CO_2$, an ocean acidification-relevant level close what is predicted globally by year 2100 (940 µatm $CO_2$ under business as usual [2]) (figure 1*a*; 1200 µatm $CO_2$). Of the two behavioural responses tested, tail withdrawal was impacted by high $CO_2$ exposure, as hypothesized, whereas righting was not (figure 2).

It has long been known that invertebrates show more inherent variation in acid–base regulatory ability than fish. Generally, active invertebrates tend to show a stronger $HCO_3^-$ buffering capacity, while less active invertebrates may experience metabolic suppression associated with a decline in pH [19,37]. In studies addressing acid–base status of invertebrates at ocean acidification-relevant $CO_2$ levels (less than approximately 2000 µatm $CO_2$), sea urchins (*Paracentrotus lividus, Echinometra mathaei, Tripneustes ventricosus*) [36,38,61–63], Arctic spider crabs (*Hyas araneus*) [35], velvet swimming crabs (*Necora puber*) [64] and shore crabs (*Carcinus maenas*) [33] all accumulate $HCO_3^-$ to correct an acidosis. In contrast, blue mussels (*Mytius edulis*) [38], king scallops (*Pecten maximus*) [34], northern sea urchins (*Strongylocentrotus drobachiensis*) [65], sea stars (*Asteria rubens, Leptasterias polaris*) [33,66], slate pencil sea urchins (*Eucidaris tribuloides*) [36] and Arctic spider crabs at higher $CO_2$ levels (3000 µatm $CO_2$) [35] show incomplete or an absence of $HCO_3^-$ accumulation that is often insufficient in maintaining pH during high $CO_2$ exposure. The diversity in acid–base responses to $CO_2$ seen among invertebrates offers a fruitful avenue for studies of the mechanistic underpinnings of disturbed behaviour. Responses in animals showing regulatory and non-regulatory responses can be studied in the same species using Aplysia. It is clear that they regulate pH at lower $CO_2$ levels (1200 µatm $CO_2$) but cannot maintain this response at higher $CO_2$ levels (3000 µatm $CO_2$) (figure 1*a*; electronic supplementary material, figure S1).

Similar to acid–base regulatory ability, the behavioural responses of invertebrates have been variable. In the present study, $CO_2$ exposure did not alter the self-righting response of Aplysia (figure 2*a*). This mirrors self-righting results of $CO_2$-exposed gastropod molluscs (*Gibberulus gibbosus*) [13] and sea stars (*Asteria rubens*) [33]. Some studies have noted a faster righting time with elevated $CO_2$, in brittlestars (*Ophiura ophiura*) at higher $CO_2$ levels (corresponding to pH 7.3) [67], and in the Chilean abalone (*Concholepas concholepas*) [68]. In one case, righting time has been shown to increase with elevated $CO_2$

exposure in a marine gastropod (*Margarella antarctica*) [69], and there was a trend of an increase in the cone snail (*Conus marmoreus*; $p = 0.052$) [70]. The tail withdrawal, a defence mechanism elicited by Aplysia, showed a significant decrease in reflex time at elevated $CO_2$ levels (figure 2*b*), taking more time to relax the tail to half its original length after maximum contraction, but showed no change to the magnitude of the response (% body length contracted; figure 2*c*). Animals exposed to elevated $CO_2$ relaxed their tail approximately 37% faster compared with control animals. The decrease in the timing of the tail-withdrawal reflex could suggest a decline in antipredator response or increased boldness, findings that have been observed across taxa [9,13,14,16]. For example, the marine snail *G. gibbosus* jumped away from a predator cue less frequently and with increased latency when exposed to elevated $CO_2$ [13]. Similarly, flight behaviour of the black turban snail (*T. funebralis*) was reduced with increasing $CO_2$, albeit at higher $CO_2$ levels corresponding to a pH of 7.1 [15].

Given the ubiquity of $CO_2$-induced behavioural disruptions across taxa, a common neural mechanism has been proposed, where altered $HCO_3^-$ and $Cl^-$ ion gradients resulting from efforts to maintain pH homeostasis are presumed to change the function of the $GABA_A$ receptor [11]. Despite the proposed link between acid–base regulatory ability and behavioural disruptions in marine organisms, this study represents one of few that have measured both parameters in the same species at ocean acidification-relevant $CO_2$ levels [30–35]. Although Aplysia experienced an acidosis at 3000 µatm $CO_2$, they were still able to accumulate $HCO_3^-$ at both $CO_2$ levels. Based on extracellular measurements, this change could alter neuronal gradients and possibly explained shortened time to tail relaxation. However, in other invertebrate studies, sea stars unable to elevate $HCO_3^-$ [32,33] and crabs able to elevate $HCO_3^-$ both showed no difference in righting [35]. Scallops showing an acidosis with very limited $HCO_3^-$ accumulation showed significant impacts on clapping performance [34]. The lack of consistency across studies and in Aplysia in the current study may seem difficult to reconcile. The source of variation could stem from a number of factors including differential intracellular pH regulation or behavioural compensatory mechanisms. In addition, these variations may reflect that certain behaviours are not GABA-mediated. It is clear that the field would benefit from more measurements of acid–base parameters in species showing behavioural disruptions to help resolve these discrepancies.

Although the involvement of the $GABA_A$ receptor was not directly tested in the present study, $GABA_A$ receptor involvement in $CO_2$-induced behavioural disruptions have been demonstrated in some fish [8,10,23–28] and invertebrates [13,29]. These studies have largely implicated $GABA_A$ receptor involvement using whole-animal exposure to pharmacological agents targeting $GABA_A$. This method has been fundamental in establishing the proposed mechanism, but lacks resolution in targeting specific mechanisms responsible for a given behavioural disturbance. Since Aplysia accumulate $HCO_3^-$ and show a significant behavioural disruption at both tested $CO_2$ levels, they are an ideal candidate for obtaining a better understanding of mechanisms underlying $CO_2$ behavioural impairment. Findings from the present study combined with decades of research examining the electrophysiological basis of learning means that methods to link well-characterized neural networks to specific behaviours are already established. For many reflexes, including the $CO_2$-impacted tail-withdrawal reflex, the reflex can be elicited in *in vitro* preparations where the specific neural network for a given reflex is isolated from the animal [56]. In addition, Aplysia neurons are large and amenable to patch clamp techniques, where individual cells and/or specific transporters can be investigated. While the specific role of the $GABA_A$ is not well-studied in the context of the $CO_2$-impacted tail-withdrawal reflex, gamma-aminobutyric acid (GABA) has been localized to certain areas in the pedal ganglia [71], which innervates the tail [42,72]. Furthermore, Aplysia neurons from a number of regions including the pleural ganglia (also involved in the tail-withdrawal reflex), have shown both excitatory and inhibitory currents with the application of GABA and were found to be reactive to $GABA_A$ receptor antagonists [73].

In summary, we believe all of the advantages of using Aplysia as a biomedical research model for learning could be applied to ocean acidification research. Aplysia meet three important criteria (1–3). In addition to simple and well-mapped nervous systems (1), there are established and reproducible behavioural assays (2) that can be applied to examine all major forms of learning including habituation, sensitization, classical conditioning and operant conditioning [74]. Most importantly, the present study demonstrates that Aplysia accumulate $HCO_3^-$ at an ocean acidification-relevant $CO_2$ level (3). These three criteria allow for further exploration of the proposed link between acid–base regulatory ability and behaviour, including detailed testing of $GABA_A$ hypothesis.

Ethics. This study was conducted on laboratory bred and reared animals and no permits or approvals were required.
Data accessibility. Our data are deposited at the Dryad Digital Repository: https://doi.org/10.5061/dryad.7pd654v [75].

**Authors' contributions.** R.L.Z. and R.M.H. designed the study. R.L.Z. and R.M.H. conducted experiments, with R.L.Z. performing and analysing behavioural assays. R.L.Z. and R.M.H. analysed acid–base findings. R.L.Z. and R.M.H. wrote and edited the manuscript.

**Competing interests.** We have no competing interests.

**Funding.** This research was financially supported by National Institute of Health Bridge to Baccalaureate Program (Award R25GM050083).

**Acknowledgements.** We would like to thank the staff and investigators at the National Resource for Aplysia (National Institutes of Health Grant P40OD010952), with special thanks to Phillip Gillette, Dustin Stommes, Nick Kron and Dr Michael Schmale. We would especially like to thank Dr Lynne Fieber for use of laboratory space and invaluable guidance throughout the project. We would also like to acknowledge Martin Grosell for providing input on early versions of the manuscript. We would finally like to thank Michael Jarrold and Lela Schlenker for statistical advice.

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
