## [Reviewer comments · Royal Society Open Science]

Review History

RSOS-191041.R0 (Original submission)

Review form: Reviewer 1

Is the manuscript scientifically sound in its present form?

Yes

Are the interpretations and conclusions justified by the results?

Yes

Is the language acceptable?

Yes

Do you have any ethical concerns with this paper?

No

Have you any concerns about statistical analyses in this paper?

No

Recommendation?

Accept with minor revision (please list in comments)

Comments to the Author(s)

This neat study shows that the model organism, *Aplysia*, is an acid-base regulator that accumulates HCO_3^- to defend pH in high CO_2 , and that ocean acidification relevant CO_2 levels also cause behavioural changes (faster relaxation of the tail retraction reflex – an anti-predation behaviour). This is the first time that HCO_3^- accumulation and specific behavioral changes at elevated CO_2 have been directly linked in a marine invertebrate. Moreover, because *Aplysia* is a model organism for studying invertebrate neurobiology, so this study shows that *Aplysia* is an ideal species in which to investigate how higher CO_2 levels affect the behaviours of marine invertebrates, and more specifically, to test the GABA hypothesis. This is an important study that has been carefully done and the data appropriately analysed. The authors have done an excellent job responding to the original round of reviews for Proc R Soc London and I have just a few relatively minor comments and suggestions on this version of the ms, which is already in very good shape.

Minor comments

Line 41. Change “expected” to “projected”, because that’s what models do.

Lines 41-42. These levels are relevant to a business as usual scenario of CO_2 emissions, not more conservative emissions scenarios, so I suggest you finish the sentence with “under a business-as-usual CO_2 emissions scenario” or “unless the rate of CO_2 emissions is substantially curtailed”

Line 57. Insert “high” or “elevated” before “ CO_2 ”. There is still CO_2 in ambient conditions.

Line 57. Insert “animals that are” before “acid-base regulators”. Otherwise a regulator could be interpreted as some kind of tissue or organ.

Line 59. Insert “acid-base” before “regulators”

Line 61 Insert “is” after “

Para starting line 79. This should not be a new paragraph. Merge with the one above.

Para starting line 86. I think this should be part of the same single para above, too. It’s all about testing the GABA hypothesis.

Line 114. Insert “elevated” before “ CO_2 ”.

Line 137. “which” should be “that” in this instance.

Line 225. Insert “treatments” or “exposure levels” after “ CO_2 ”

Line 228. It is not clear to me why you wanted to use post-hoc testing with multiple comparisons, which requires you to use an adjusted p-value. Aren’t you simply interested in how each of the high CO_2 treatments compare with the control? In that case you can simply use the contrast outputted by the LME (i.e. each treatment compared with the control) and you don’t need post-hoc tests that are then adjusted for multiple comparisons. All you need do is make sure the control is coded in a way that it is selected by the model as the contrast treatment. You only need the post-hoc tested if you specifically want to compare ALL the treatment levels to each other. None of this may matter because of the strength of your results, but thought it should be pointed out.

Line 263. Replace “the time it took to right” with “time to right”.

Line 270. Insert “high” before “ CO_2 ”. This seems like a redundant analysis to me.

Line 281. Insert “high” before “ CO_2 ”.

Line 296. Insert “high” before “ CO_2 ”.

Line 298. Not clear how a species can be both a regulator and non-regulator. I thought you were arguing that acid-base regulation is a species level trait. Do you mean by using different species of *Aplysia*? Some clarification needed here.

Line 309. "and" in the Chilean abalone.

311. I don't think you need "interestingly" here.

312. Insert "elevated" before "CO₂"

314 Delete "in other words"

Line 333. This is a poorly explained/constructed sentence. I think you mean there was no difference in the righting response both in crabs that are able to elevated HCO₃⁻ and sea stars that are unable to elevated HCO₃⁻.

Line 340. You say: "In marine fish, the link between regulatory ability and behaviour has been more consistent, where HCO₃⁻ accumulation has been linked to impaired olfaction and lateralization [29, 30]." The problem is that HCO₃⁻ accumulation has not been tested in fish species that do NOT show behavioural changes in high CO₂, so we do not know if there is actually a good correlation here or not. It's drawing a very long bow to say it's consistent. I suggest you delete this sentence.

Line 347. Tested in less than ten species, so not "many" species, really. Better to say some fish and invertebrates.

Review form: Reviewer 2

Is the manuscript scientifically sound in its present form?

Yes

Are the interpretations and conclusions justified by the results?

Yes

Is the language acceptable?

Yes

Do you have any ethical concerns with this paper?

No

Have you any concerns about statistical analyses in this paper?

No

Recommendation?

Accept with minor revision (please list in comments)

Comments to the Author(s)

Zlatkin & Heuer have done a thorough job addressing the reviewer comments from a previous submission of the manuscript to another journal.

I have only minor comments in this review.

L41-43 - References 1 and 2 are old now. Cite IPCC reports or similar for more recent CO₂ projections.

Given that “Animals were exposed for either 4 or 11 days to each CO₂ level” as stated in the methods, I recommend the authors change “4-11 days” to “4 or 11 days” in the manuscript and figure legends.

Behavioral count style data – i.e. time to right and time to relax tail – would be better analyzed on raw data with a count style distribution (e.g. Poisson, etc) with a GLMM. However, looking at the graphs, it is likely that the conclusions from the current manuscript tests (using an LME with log-transformation) would be the same.

L269 – there is a typo mistake in the P value reported as “P=0.0.027”.

L280 – the reference to Figure 1a; 1000 μ atm CO₂ is unclear or incorrect. Figure 1a has 400, 1200 and 3000 μ atm as pCO₂ treatment conditions. Further, global year 2100 CO₂ projections are 940 ppm (RCP 8.5) or less – please change this sentence.

Figure 1 and 2 – change the pCO₂ treatment conditions all to 400, 1200 and 3000 μ atm. At present there is a mixture of 1000 and 1200 μ atm reported for the mid-CO₂ level on both these figures, however, the Supp Table 2 shows these values should all be 1200.

In Supp Table 2, TA increases with pCO₂. This is unusual, as TA should remain constant while pCO₂ and TCO₂ change in CO₂ manipulation studies. Otherwise there is a confounding effect of TA and pCO₂ (i.e. TA changes at the same time as pCO₂, but the authors are interpreting the results in terms of pCO₂). Please can the authors detail why TA changes with pCO₂ in their experiments.

Fig S1 - state the errors presented.

This manuscript does not test GABA, so I recommend to remove GABA mentions from key words to avoid misleading readers.

L323-326 – this sentence needs references, since the proposal of this neural mechanism is not the authors’ own.

Manuscript needs proof-reading.

L374 – note that the pCO₂ levels at 1200 μ atm in this study exceeds ‘near-future CO₂ levels’, which are usually considered as those at the end of this century. I would prefer ‘near-future CO₂ level’ statements to be removed from the manuscript.

Decision letter (RSOS-191041.R0)

22-Aug-2019

Dear Dr Heuer

On behalf of the Editors, I am pleased to inform you that your Manuscript RSOS-191041 entitled "Ocean acidification affects acid-base physiology and behaviour in a model invertebrate, the California sea hare (*Aplysia californica*)" has been accepted for publication in Royal Society Open Science subject to minor revision in accordance with the referee suggestions. Please find the referees' comments at the end of this email.

The reviewers and handling editors have recommended publication, but also suggest some minor revisions to your manuscript. Therefore, I invite you to respond to the comments and revise your manuscript.

- Ethics statement

- Data accessibility

If you wish to submit your supporting data or code to Dryad (<http://datadryad.org/>), or modify your current submission to dryad, please use the following link:
<http://datadryad.org/submit?journalID=RSOS&manu=RSOS-191041>

- Competing interests

- Authors' contributions

- Acknowledgements

- Funding statement

Because the schedule for publication is very tight, it is a condition of publication that you submit the revised version of your manuscript before 31-Aug-2019. Please note that the revision deadline will expire at 00.00am on this date. If you do not think you will be able to meet this date please let me know immediately.

on behalf of Kevin Padian (Subject Editor)
openscience@royalsociety.org

Reviewer comments to Author:

Reviewer: 1

This neat study shows that the model organism, *Aplysia*, is an acid-base regulator that accumulates HCO_3^- to defend pH in high CO_2 , and that ocean acidification relevant CO_2 levels also cause behavioural changes (faster relaxation of the tail retraction reflex – an anti-predation behaviour). This is the first time that HCO_3^- accumulation and specific behavioral changes at elevated CO_2 have been directly linked in a marine invertebrate. Moreover, because *Aplysia* is a model organism for studying invertebrate neurobiology, so this study shows that *Aplysia* is an ideal species in which to investigate how higher CO_2 levels affect the behaviours of marine invertebrates, and more specifically, to test the GABA hypothesis. This is an important study that has been carefully done and the data appropriately analysed. The authors have done an excellent job responding to the original round of reviews for Proc R Soc London and I have just a few relatively minor comments and suggestions on this version of the ms, which is already in very good shape.

Minor comments

Line 41. Change “expected” to “projected”, because that’s what models do.

Lines 41-42. These levels are relevant to a business as usual scenario of CO_2 emissions, not more conservative emissions scenarios, so I suggest you finish the sentence with “under a business-as-usual CO_2 emissions scenario” or “unless the rate of CO_2 emissions is substantially curtailed”

Line 57. Insert “high” or “elevated” before “ CO_2 ”. There is still CO_2 in ambient conditions.

Line 57. Insert “animals that are” before “acid-base regulators”. Otherwise a regulator could be interpreted as some kind of tissue or organ.

Line 59. Insert “acid-base” before “regulators”

Line 61 Insert “is” after “

Para starting line 79. This should not be a new paragraph. Merge with the one above.

Para starting line 86. I think this should be part of the same single para above, too. It's all about testing the GABA hypothesis.

Line 114. Insert "elevated" before "CO₂".

Line 137. "which" should be "that" in this instance.

Line 225. Insert "treatments" or "exposure levels" after "CO₂"

Line 228. It is not clear to me why you wanted to use post-hoc testing with multiple comparisons, which requires you to use an adjusted p-value. Aren't you simply interested in how each of the high CO₂ treatments compare with the control? In that case you can simply use the contrast outputted by the LME (i.e. each treatment compared with the control) and you don't need post-hoc tests that are then adjusted for multiple comparisons. All you need do is make sure the control is coded in a way that it is selected by the model as the contrast treatment. You only need the post-hoc tested if you specifically want to compare ALL the treatment levels to each other. None of this may matter because of the strength of your results, but thought it should be pointed out.

Line 263. Replace "the time it took to right" with "time to right".

Line 270. Insert "high" before "CO₂". This seems like a redundant analysis to me.

Line 281. Insert "high" before "CO₂".

Line 296. Insert "high" before "CO₂".

Line 298. Not clear how a species can be both a regulator and non-regulator. I thought you were arguing that acid-base regulation is a species level trait. Do you mean by using different species of *Aplysia*? Some clarification needed here.

Line 309. "and" in the Chilean abalone.

311. I don't think you need "interestingly" here.

312. Insert "elevated" before "CO₂"

314 Delete "in other words"

Line 333. This is a poorly explained/constructed sentence. I think you mean there was no difference in the righting response both in crabs that are able to elevated HCO₃⁻ and sea stars that are unable to elevated HCO₃⁻.

Line 340. You say: "In marine fish, the link between regulatory ability and behaviour has been more consistent, where HCO₃⁻ accumulation has been linked to impaired olfaction and lateralization [29, 30]." The problem is that HCO₃⁻ accumulation has not been tested in fish species that do NOT show behavioural changes in high CO₂, so we do not know if there is actually a good correlation here or not. It's drawing a very long bow to say it's consistent. I suggest you delete this sentence.

Line 347. Tested in less than ten species, so not "many" species, really. Better to say some fish and invertebrates.

Reviewer: 2

Comments to the Author(s)

Zlatkin & Heuer have done a thorough job addressing the reviewer comments from a previous submission of the manuscript to another journal.

I have only minor comments in this review.

L41-43 – References 1 and 2 are old now. Cite IPCC reports or similar for more recent CO₂ projections.

Given that "Animals were exposed for either 4 or 11 days to each CO₂ level" as stated in the methods, I recommend the authors change "4-11 days" to "4 or 11 days" in the manuscript and figure legends.

Behavioral count style data – i.e. time to right and time to relax tail – would be better analyzed on raw data with a count style distribution (e.g. Poisson, etc) with a GLMM. However, looking at the graphs, it is likely that the conclusions from the current manuscript tests (using an LME with log-transformation) would be the same.

L269 – there is a typo mistake in the P value reported as “P=0.0.027”.

L280 – the reference to Figure 1a; 1000 μatm CO₂ is unclear or incorrect. Figure 1a has 400, 1200 and 3000 μatm as pCO₂ treatment conditions. Further, global year 2100 CO₂ projections are 940 ppm (RCP 8.5) or less – please change this sentence.

Figure 1 and 2 – change the pCO₂ treatment conditions all to 400, 1200 and 3000 μatm . At present there is a mixture of 1000 and 1200 μatm reported for the mid-CO₂ level on both these figures, however, the Supp Table 2 shows these values should all be 1200.

In Supp Table 2, TA increases with pCO₂. This is unusual, as TA should remain constant while pCO₂ and TCO₂ change in CO₂ manipulation studies. Otherwise there is a confounding effect of TA and pCO₂ (i.e. TA changes at the same time as pCO₂, but the authors are interpreting the results in terms of pCO₂). Please can the authors detail why TA changes with pCO₂ in their experiments.

Fig S1 - state the errors presented.

This manuscript does not test GABA, so I recommend to remove GABA mentions from key words to avoid misleading readers.

L323-326 – this sentence needs references, since the proposal of this neural mechanism is not the authors' own.

Manuscript needs proof-reading.

L374 – note that the pCO₂ levels at 1200 μatm in this study exceeds 'near-future CO₂ levels', which are usually considered as those at the end of this century. I would prefer 'near-future CO₂ level' statements to be removed from the manuscript.

Author's Response to Decision Letter for (RSOS-191041.R0)

See Appendix A.

Decision letter (RSOS-191041.R1)

06-Sep-2019

Dear Dr Heuer,

I am pleased to inform you that your manuscript entitled "Ocean acidification affects acid-base

physiology and behaviour in a model invertebrate, the California sea hare (*Aplysia californica*)" is now accepted for publication in Royal Society Open Science.

on behalf of Mr Andrew Dunn (Associate Editor) and Kevin Padian (Subject Editor)
openscience@royalsociety.org

Associate Editor Comments to Author (Mr Andrew Dunn):
Associate Editor: 1
Comments to the Author:
(There are no comments.)

Reviewer comments to Author:

Appendix A

22-Aug-2019

Dear Dr Heuer

On behalf of the Editors, I am pleased to inform you that your Manuscript RSOS-191041 entitled "Ocean acidification affects acid-base physiology and behaviour in a model invertebrate, the California sea hare (*Aplysia californica*)" has been accepted for publication in Royal Society Open Science subject to minor revision in accordance with the referee suggestions. Please find the referees' comments at the end of this email.

The reviewers and handling editors have recommended publication, but also suggest some minor revisions to your manuscript. Therefore, I invite you to respond to the comments and revise your manuscript.

- Ethics statement

- Data accessibility

If you wish to submit your supporting data or code to Dryad

(<https://nam01.safelinks.protection.outlook.com/?url=http%3A%2F%2Fdatadryad.org%2F&data=02%7C01%7Crheuer%40rsmas.miami.edu%7Cc1da00bdd6934bd7dca308d726e122b3%7C2a144b72f23942d48c0e6f0f17c48e33%7C0%7C0%7C637020620728940911&data=ytja9HC07spVGyae3O4s8zFnOTx0rcwEc2ZbpkMJ6UE%3D&reserved=0>), or modify your current submission to dryad, please use

the following link:

<https://nam01.safelinks.protection.outlook.com/?url=http%3A%2F%2Fdatadryad.org%2Fsubmit%3FjournalID%3DRSOS%26manu%3DRSOS-191041&data=02%7C01%7Crheuer%40rsmas.miami.edu%7Cc1da00bdd6934bd7dca308d726e122b3%7C2a144b72f23942d48c0e6f0f17c48e33%7C0%7C0%7C637020620728940911&data=8xRfp5K5h2%2BieO1PtG9KmrjaawQRpzauwEaWhIXCnI0%3D&reserved=0>

- Competing interests

- Authors' contributions

All submissions, other than those with a single author, must include an Authors' Contributions section which individually lists the specific contribution of each author. The list of Authors should meet all of the

following criteria; 1) substantial contributions to conception and design, or acquisition of data, or analysis and interpretation of data; 2) drafting the article or revising it critically for important intellectual content; and 3) final approval of the version to be published.

- Acknowledgements

- Funding statement

Please ensure you have prepared your revision in accordance with the guidance at

[https://nam01.safelinks.protection.outlook.com/?url=https%3A%2F%2Froyalsociety.org%2Fjournals%2Fauthors%2Fauthor-](https://nam01.safelinks.protection.outlook.com/?url=https%3A%2F%2Froyalsociety.org%2Fjournals%2Fauthors%2Fauthor-guidelines%2F&data=02%7C01%7Crheuer%40rsmas.miami.edu%7Cc1da00bdd6934bd7dca308d726e122b3%7C2a144b72f23942d48c0e6f0f17c48e33%7C0%7C0%7C637020620728940911&reserved=0)

[guidelines%2F&data=02%7C01%7Crheuer%40rsmas.miami.edu%7Cc1da00bdd6934bd7dca308d726e122b3%7C2a144b72f23942d48c0e6f0f17c48e33%7C0%7C0%7C637020620728940911&reserved=0](https://nam01.safelinks.protection.outlook.com/?url=https%3A%2F%2Froyalsociety.org%2Fjournals%2Fauthor-guidelines%2F&data=02%7C01%7Crheuer%40rsmas.miami.edu%7Cc1da00bdd6934bd7dca308d726e122b3%7C2a144b72f23942d48c0e6f0f17c48e33%7C0%7C0%7C637020620728940911&reserved=0) -- please note that we

cannot publish your manuscript without the end statements. We have included a screenshot example of the end statements for reference. If you feel that a given heading is not relevant to your paper, please nevertheless include the heading and explicitly state that it is not relevant to your work.

Because the schedule for publication is very tight, it is a condition of publication that you submit the revised version of your manuscript before 31-Aug-2019. Please note that the revision deadline will expire at 00.00am on this date. If you do not think you will be able to meet this date please let me know immediately.

To revise your manuscript, log into

<https://nam01.safelinks.protection.outlook.com/?url=https%3A%2F%2Fmc.manuscriptcentral.com%2Frosos&data=02%7C01%7Crheuer%40rsmas.miami.edu%7Cc1da00bdd6934bd7dca308d726e122b3%7C2a144b72f23942d48c0e6f0f17c48e33%7C0%7C0%7C637020620728940911&reserved=0> and enter your Author Centre,

where you will find your manuscript title listed under "Manuscripts with Decisions". Under "Actions," click on "Create a Revision." You will be unable to make your revisions on the originally submitted version of the manuscript. Instead, revise your manuscript and upload a new version through your Author Centre.

Supplementary files will be published alongside the paper on the journal website and posted on the online figshare repository

(<https://nam01.safelinks.protection.outlook.com/?url=https%3A%2F%2Frs.figshare.com%2F&data=02%7C01%7Crheuer%40rsmas.miami.edu%7Cc1da00bdd6934bd7dca308d726e122b3%7C2a144b72f23942d48c0e6f0f17c48e33%7C0%7C0%7C637020620728940911&data=i3KgKyfgrGZjj%2F6RQoPdQ9hrVBFZgkq%2F9YYpNte4VV4%3D&reserved=0>). The heading and legend provided for each supplementary file during the submission process will be used to create the figshare page, so please ensure these are accurate and informative so that your files can be found in searches. Files on figshare will be made available approximately one week before the accompanying article so that the supplementary material can be attributed a unique DOI.

Please note that Royal Society Open Science charge article processing charges for all new submissions that are accepted for publication. Charges will also apply to papers transferred to Royal Society Open Science from other Royal Society Publishing journals, as well as papers submitted as part of our collaboration with the Royal Society of Chemistry

(<https://nam01.safelinks.protection.outlook.com/?url=http%3A%2F%2Frsos.royalsocietypublishing.org%2Fchemistry&data=02%7C01%7Crheuer%40rsmas.miami.edu%7Cc1da00bdd6934bd7dca308d726e122b3%7C2a144b72f23942d48c0e6f0f17c48e33%7C0%7C0%7C637020620728940911&data=mCSRAqozuFD4JqB9g4Zbu5EnR3ZsOtGFNGjLgE40Ok%3D&reserved=0>).

If your manuscript is newly submitted and subsequently accepted for publication, you will be asked to pay the article processing charge, unless you request a waiver and this is approved by Royal Society Publishing. You can find out more about the charges at

<https://nam01.safelinks.protection.outlook.com/?url=http%3A%2F%2Frsos.royalsocietypublishing.org%2Fpage%2Fcharges&data=02%7C01%7Crheuer%40rsmas.miami.edu%7Cc1da00bdd6934bd7dca308d726e122b3%7C2a144b72f23942d48c0e6f0f17c48e33%7C0%7C0%7C637020620728940911&data=>

ta=qGg9YtM8SIDboXmz38wBdcrwsWZX1TRwyzfcFyAmrxk%3D&reserved=0. Should you have any queries, please contact openscience@royalsociety.org.

on behalf of Kevin Padian (Subject Editor) openscience@royalsociety.org

Reviewer comments to Author:

Reviewer: 1

This neat study shows that the model organism, Aplysia, is an acid-base regulator that accumulates HCO₃⁻ to defend pH in high CO₂, and that ocean acidification relevant CO₂ levels also cause behavioural changes (faster relaxation of the tail retraction reflex – an anti-predation behaviour). This is the first time that HCO₃⁻ accumulation and specific behavioral changes at elevated CO₂ have been directly linked in a marine invertebrate. Moreover, because Aplysia is a model organism for studying invertebrate neurobiology, so this study shows that Aplysia is an ideal species in which to investigate how higher CO₂ levels affect the behaviours of marine invertebrates, and more specifically, to test the GABA hypothesis. This is an important study that has been carefully done and the data appropriately analysed. The authors have done an excellent job responding to the original round of reviews for Proc R Soc London and I have just a few relatively minor comments and suggestions on this version of the ms, which is already in very good shape.

We thank the reviewer for their kind comments and thorough review of the manuscript.

Minor comments

Line 41. Change “expected” to “projected”, because that’s what models do.

We have changed this as requested. L41-43

“Average global oceanic CO₂ levels are projected to increase from current levels of ~400 to ~940 μatm CO₂ by the end of the century and ~1900 μatm CO₂ by the year 2300 unless the rate of CO₂ emissions is substantially curtailed.”

Lines 41-42. These levels are relevant to a business as usual scenario of CO₂ emissions, not more conservative emissions scenarios, so I suggest you finish the sentence with “under a business-as-usual CO₂ emissions scenario” or “unless the rate of CO₂ emissions is substantially curtailed”

We have altered the wording and appreciate the alternative suggestions. The sentence now reads as follows L41-43:

“Average global oceanic CO₂ levels are projected to increase from current levels of ~400 to ~940 μatm CO₂ by the end of the century and ~1900 μatm CO₂ by the year 2300 unless the rate of CO₂ emissions is substantially curtailed.”

Line 57. Insert “high” or “elevated” before “CO₂”. There is still CO₂ in ambient conditions.

We have added the word “elevated” as suggested. L58

Line 57. Insert “animals that are” before “acid-base regulators”. Otherwise a regulator could be interpreted as some kind of tissue or organ.

Good point, we have added this to the sentence as suggested. L58-61

“Following the onset of CO₂ exposure, animals that are acid-base “regulators” counter an initial drop in blood pH through the retention and/or uptake of HCO₃⁻. This process allows acid-base regulators to correct...”

Line 59. Insert “acid-base” before “regulators”

We have added this to the sentence as suggested. L60

Line 61 Insert “is” after “

We have altered the sentence as follows: L61-62

“This compensation mechanism is generally related to how “active” an organism is, as higher metabolic rates (O₂ consumption) necessitate higher rates of CO₂ excretion.”

Para starting line 79. This should not be a new paragraph. Merge with the one above.

We have merged the paragraphs as suggested.

Para starting line 86. I think this should be part of the same single para above, too. It’s all about testing the GABA hypothesis.

We have merged the paragraphs as suggested.

Line 114. Insert “elevated” before “CO₂”.

We have added this to the sentence as suggested. L114

Line 137. “which” should be “that” in this instance.

We have changed this as suggested. L140-141

“Animals that experienced 11-day exposures were subjected to the same ~96-hour fasting period.”

Line 225. Insert “treatments” or “exposure levels” after “CO₂”

We have added “exposure levels” as requested. L228-229.

“Linear mixed effect (LME) models were used to test for responses to CO₂ exposure levels for the time to complete the righting reflex and the time to complete the tail-withdrawal reflex.”

Line 228. It is not clear to me why you wanted to use post-hoc testing with multiple comparisons, which requires you to use an adjusted p-value. Aren't you simply interested in how each of the high CO₂ treatments compare with the control? In that case you can simply use the contrast outputted by the LME (i.e. each treatment compared with the control) and you don't need post-hoc tests that are then adjusted for multiple comparisons. All you need do is make sure the control is coded in a way that it is selected by the model as the contrast treatment. You only need the post-hoc tested if you specifically want to compare ALL the treatment levels to each other. None of this may matter because of the strength of your results, but thought it should be pointed out.

We decided to use the post-hoc comparisons to address a comment from a reviewer in a previous submission regarding correcting data for multiple comparisons. In addition, we were interested in the pairwise comparison between 1200 and 3000 μ atm. After checking our model, we note that using the model output or the pairwise comparisons does not change the strength of the results.

Line 263. Replace “the time it took to right” with “time to right”.

We have changed this as suggested. L265-266

Line 270. Insert “high” before “CO₂”. This seems like a redundant analysis to me.

We have added “high” as suggested (L273). We have addressed the second portion of this comment in the Line 228 response above.

Line 281. Insert “high” before “CO₂”.

We have added “high” as suggested. L284

Line 296. Insert “high” before “CO₂”.

We have added “high” as suggested. L299

Line 298. Not clear how a species can be both a regulator and non-regulator. I thought you were arguing that acid-base regulation is a species level trait. Do you mean by using different species of Aplysia? Some clarification needed here.

We agree the wording may be confusing. Theoretically, any animal could change from acid-base regulating to non-regulating if exposed to high enough CO₂. Rather, we are referring here to regulatory and non-regulatory responses. We have changed the sentence to reflect this detail.

L301-302: “Responses in animals showing regulatory and non-regulatory responses can be studied in the same species using Aplysia.”

Line 309. “and” in the Chilean abalone.

We have added “and” as suggested. L310

311. I don't think you need “interestingly” here.

We have deleted this as requested.

312. Insert “elevated” before “CO2”

We have changed the sentence as suggested. L312

314 Delete “in other words”

We have made the deletion as requested.

Line 333. This is a poorly explained/constructed sentence. I think you mean there was no difference in the righting response both in crabs that are able to elevated HCO₃⁻ and sea stars that are unable to elevated HCO₃⁻.

We agree that this sentence needs more clarity. We have revised it as follows:

L335-337: However, in other invertebrate studies, sea stars unable to elevate HCO₃⁻ (Appelhans et al. 2014; Appelhans et al. 2012) and crabs able to elevate HCO₃⁻ both showed no difference in righting (Zittier et al. 2013).

Line 340. You say: “In marine fish, the link between regulatory ability and behaviour has been more consistent, where HCO₃⁻ accumulation has been linked to impaired olfaction and lateralization [29, 30].” The problem is that HCO₃⁻ accumulation has not been tested in fish species that do NOT show behavioural changes in high CO₂, so we do not know if there is actually a good correlation here or not. It's drawing a very long bow to say it's consistent. I suggest you delete this sentence.

We have deleted this sentence as suggested.

Line 347. Tested in less than ten species, so not “many” species, really. Better to say some fish and invertebrates.

We have changed “many” to “some” as suggested. L347

Reviewer: 2

Comments to the Author(s)

Zlatkin & Heuer have done a thorough job addressing the reviewer comments from a previous submission of the manuscript to another journal.

We appreciate the reviewer's summary of our revisions and for their comments.

I have only minor comments in this review.

L41-43 – References 1 and 2 are old now. Cite IPCC reports or similar for more recent CO2 projections.

We have removed the Meehl citation and added two more recent citations including the most recent IPCC report.

(Meinshausen et al. 2011; Portner et al. 2014)

Given that “Animals were exposed for either 4 or 11 days to each CO2 level” as stated in the methods, I recommend the authors change “4-11 days” to “4 or 11 days” in the manuscript and figure legends.

We included the 4 or 11 days to be transparent about our methods. We observed no effect of the day of exposure in any our statistical models, so we pooled the data across days and using 4-11 days throughout the rest of the manuscript, as suggested by a previous reviewer. We have added the following sentence in the methods, where this issue is first presented to avoid confusion. L133-135

“Since day of exposure (4 versus 11) did not significantly impact any measured endpoint (see below), exposure duration is referred to as 4-11 days throughout the manuscript.”

Behavioral count style data – i.e. time to right and time to relax tail – would be better analyzed on raw data with a count style distribution (e.g. Poisson, etc) with a GLMM. However, looking at the graphs, it is likely that the conclusions from the current manuscript tests (using an LME with log-transformation) would be the same.

We agree and would prefer to stick with the current tests provided in the manuscript.

L269 – there is a typo mistake in the P value reported as “P=0.0.027”.

We have fixed this error, thank you. L268

L280 – the reference to Figure 1a; 1000 μatm CO2 is unclear or incorrect. Figure 1a has 400, 1200 and 3000 μatm as pCO2 treatment conditions. Further, global year 2100 CO2 projections are 940 ppm (RCP 8.5) or less – please change this sentence.

We have changed the projected level to 940, and corrected the error in both the figures and the text. The correct value is 1200 μatm CO₂. This sentence now reads as follows: L281-283

“This compensatory effort led to complete pH defense at 1200 μatm CO₂, an ocean acidification relevant level close to what is predicted globally by year 2100 (940 μatm CO₂ under business as usual [2]) (Figure 1a; 1200 μatm CO₂).”

Figure 1 and 2 – change the pCO2 treatment conditions all to 400, 1200 and 3000 μatm . At present there is a mixture of 1000 and 1200 μatm reported for the mid-CO2 level on both these figures, however, the Supp Table 2 shows these values should all be 1200.

The correct value is 1200 μatm CO₂ as noted by the reviewer. We have fixed this error in the figures.

In Supp Table 2, TA increases with pCO2. This is unusual, as TA should remain constant while pCO2 and TCO2 change in CO2 manipulation studies. Otherwise there is a confounding effect of TA and

pCO₂ (i.e. TA changes at the same time as pCO₂, but the authors are interpreting the results in terms of pCO₂). Please can the authors detail why TA changes with pCO₂ in their experiments.

We examined the data to see if there was a significant difference between the TA values for each group. We ran three one-way ANOVAs to test for treatment effects on acid-base water chemistry TA, tail-withdrawal water chemistry TA, and righting water chemistry TA. Although TA values appear to be increasing, none of these increases are statistically significant across CO₂ levels. We also calculated that the percent increase from control to high CO₂ (at either CO₂ level), and this number was relatively low: 4.4-6.6%. This is relatively small compared to the large changes noted in pCO₂ across levels.

Fig S1 - state the errors presented.

We have added the following to the figure caption.

“Error bars represent SEM.”

This manuscript does not test GABA, so I recommend to remove GABA mentions from key words to avoid misleading readers.

We have removed mention of gaba in the keywords.

L323-326 – this sentence needs references, since the proposal of this neural mechanism is not the authors’ own.

We have added Nilsson et al 2012 as a reference for this sentence. L326-329

Manuscript needs proof-reading.

We have re-read this carefully before final submission, and corrected the errors pointed out in earlier comments.

L374 – note that the pCO₂ levels at 1200 µatm in this study exceeds ‘near-future CO₂ levels’, which are usually considered as those at the end of this century. I would prefer ‘near-future CO₂ level’ statements to be removed from the manuscript.

We have removed “near-future” and replaced it with “ocean acidification-relevant.” L373-374.

“Most importantly, the present study demonstrates that *Aplysia* accumulate HCO₃⁻ at an ocean acidification-relevant CO₂ level.”

Journal Name: Royal Society Open Science Journal Code: RSOS Online ISSN: 2054-5703 Journal Admin Email: openscience@royalsociety.org Journal Editor: Andrew Dunn Journal Editor Email: openscience@royalsociety.org MS Reference Number: RSOS-191041 Article Status: SUBMITTED MS Dryad ID: RSOS-191041 MS Title: Ocean acidification affects acid-base physiology and behaviour in a model invertebrate, the California sea hare (*Aplysia californica*) MS Authors: Zlatkin, Rebecca; Heuer, Rachael Contact Author: Rachael Heuer Contact Author Email: rheuer@rsmas.miami.edu Contact Author Address 1: 4600 Rickenbacker Causeway Contact Author Address 2:

Contact Author Address 3:

Contact Author City: Miami

Contact Author State: Florida

Contact Author Country: United States

Contact Author ZIP/Postal Code: 33149

Keywords: GABA, CO₂, mollusc, carbon dioxide, climate change, GABA_A receptor

Abstract: Behavioural impairment following exposure to ocean acidification-relevant CO₂ levels has been noted in a broad array of taxa. The underlying cause of these disruptions is thought to stem from alterations of ion gradients (HCO₃⁻/Cl⁻) across neuronal cell membranes that occur as a consequence of maintaining pH homeostasis via the accumulation of HCO₃⁻/Cl⁻. While behavioural impacts are widely documented, few studies have measured acid-base parameters in species showing behavioural disruptions. In addition, current studies examining mechanisms lack resolution in targeting specific neural pathways corresponding to a given behaviour. With these considerations in mind, acid-base parameters and behaviour were measured in a model organism utilized for decades as a research model to study learning, the California sea hare (*Aplysia californica*). *Aplysia* exposed to CO₂ elevated hemolymph HCO₃⁻/Cl⁻, achieving full and partial pH compensation at 1200 and 3000 µatm CO₂, respectively. Increased CO₂ did not affect self-righting behaviour. In contrast, both levels of elevated CO₂ reduced the time of the tail-withdrawal reflex, suggesting a reduction in antipredator response. Overall, these results confirm that *Aplysia* are promising models to examine mechanisms underlying CO₂-induced behavioural disruptions since they regulate HCO₃⁻/Cl⁻ and have behaviours linked to neural networks amenable to electrophysiological testing.

EndDryadContent

References for response to reviewer comments:

Appelhans, Y.S., J. Thomsen, S. Opitz, C. Pansch, F. Melzner and M. Wahl. 2014. Juvenile sea stars exposed to acidification decrease feeding and growth with no acclimation potential. *Marine Ecology Progress Series* 509: 227-239.

Appelhans, Y.S., J. Thomsen, C. Pansch, F. Melzner and M. Wahl. 2012. Sour times: seawater acidification effects on growth, feeding behaviour and acid–base status of *Asterias rubens* and *Carcinus maenas*. *Marine Ecology Progress Series* 459: 85-98.

Meinshausen, M., S.J. Smith, K. Calvin, J.S. Daniel, M. Kainuma, J. Lamarque, K. Matsumoto, S. Montzka, S. Raper and K. Riahi. 2011. The RCP greenhouse gas concentrations and their extensions from 1765 to 2300. *Climatic Change* 109: 213-241.

Portner, H.O., D.M. Karl, P.W. Boyd, S.E. Cheung, S.E. Lluch-Cota, Y. Nojiri, D.N. Schmidt and P.O. Zavialov. 2014. Ocean systems. pp. 411-484 in C.B. Field, V.R. Barros, D.J. Dokken, K.J. Mach, M.D. Mastrandrea, T.E. Bilir, M. Chatterjee, K.L. Ebi, Y.O. Estrada, R.C. Genova, B. Girma, E.S. Kissel, A.N. Levy, S. MacCracken, P.R. Mastrandrea and L.L. White, eds. *Climate Change 2014: Impacts, Adaptation, and Vulnerability. Part A: Global and Sectoral Aspects. Contribution of Working Group II to the Fifth Assessment Report of the Intergovernmental Panel on Climate Change*. Cambridge University Press, Cambridge, U.K. and New York, NY, USA.

Zittier, Z.M., T. Hirse and H.-O. Pörtner. 2013. The synergistic effects of increasing temperature and CO₂ levels on activity capacity and acid–base balance in the spider crab, *Hyas araneus*. *Marine biology* 160: 2049-2062.